# Social information decreases giving in late-stage fundraising campaigns

**Coby Morvinski**[1]*, **Matthew J. Lupoli**[2], **On Amir**[3]

**1** Department of Management, Ben-Gurion University, Be'er Sheva, Israel, **2** Department of Management, Monash University, Caulfield East, Australia, **3** Department of Marketing, UC San Diego, San Diego, California, United States of America

\* cobym@bgu.ac.il

**Data Availability Statement:** Study 1 data was provided directly by JGive with their permission to publish research findings and remained within their terms and conditions. Data, analysis scripts, and survey materials for Studies 2 and 3 are available

## Abstract

Online fundraisers often showcase information about the number of donations received and the proximity to the campaign goal. This practice follows research on descriptive norms and goal-directed motivation, which predicts higher contributions as the number of donors increases and as the campaign goal is approached. However, across three studies, we demonstrate that when the campaign is close to completion, individuals give more when they see that there are few (vs. many) donors to the campaign. We observe this result across real campaigns on a fundraising website and obtain causal evidence for this effect in two laboratory experiments. We find that this effect is driven in part by an increase in the perceived progress that one's donation makes towards reaching the campaign goal. This work identifies a counterintuitive consequence of norm-based marketing appeals and has important implications for fundraisers.

## Introduction

Online fundraising for philanthropic causes is becoming an increasingly popular method to tackle humanity's most pressing problems. With the rising prevalence of online fundraisers, it is common for fundraising websites to present live, up-to-date giving metrics for hosted campaigns to potential donors. For example, many online fundraising platforms display information such as the number of donors to the campaign and the percentage of the campaign goal reached to potential donors. Perhaps one reason why fundraisers provide this information is because they believe that people will give more as the campaign grows in popularity and support (i.e., social proof), as measured by the number of donors and progress towards the campaign goal. But given that only 22.4% of campaigns globally reach their funding goals [1], is this assumption correct?

Research on descriptive norms suggests that fundraisers would be wise to provide social information about the number of donors to a campaign, especially when many people have donated. Information about descriptive social norms, or what other people do, has been shown to influence prosocial behavior across a wide array of domains, including charitable giving [2–5]. Likewise, offering information about goal progress could also increase giving. Research on the goal gradient effect finds that people work harder to achieve goals as those

**Funding:** CM Grant #766/19 Israel Science
Foundation (ISF) www.isf.org.il The funders had no
role in study design, data collection and analysis,
decision to publish, or preparation of the
manuscript.

**Competing interests:** The authors have declared
that no competing interests exist.

goals are approached [6–8]; this effect has been identified in the context of charitable giving as well [9]. Together, this past work suggests that fundraisers would benefit from indicating to potential donors that many (as opposed to few) individuals have already donated and this benefit would further increase when the campaign goal is close to being reached.

However, we argue that these two forms of information can have counterintuitive effects on giving when presented simultaneously. Specifically, we propose that when the goal of a campaign is close to being reached, individuals may give more, and exhibit a higher likelihood of giving, when there are *few* rather than many donors. When goals are approached, each additional dollar donated toward the goal reduces a greater proportion of the remaining distance to the goal [10]. For example, a $1 donation to a campaign with a $100 goal and $0 raised previously decreases the distance to the goal by 1%; a $1 donation to that same campaign with $99 raised previously reduces the distance to the goal by 100%. Of course, the absolute (dollar) amount of goal progress made is the same in both cases. As such, a central mechanism underlying goal-directed motivation is the perception that one's effort will result in greater relative progress towards the goal [7, 9, 11]. Similarly, we expect that holding donation amount constant, giving to a nearly completed campaign with few (vs. many) donors may lead to the perception that one's donation will make a greater marginal impact on goal progress, which consequently increases motivation to give. Put differently, when there are few (vs. many) donors, each subsequent donation may be perceived as carrying greater responsibility for making progress towards the (nearly completed) campaign goal. This notion parallels findings from research on social loafing, which suggest that individuals put forth more effort to achieve a common goal when they are part of a smaller (vs. larger) group because of the perception that each individual is responsible for a greater impact on the goal [12–14]. Given this increase in the perceived goal progress made by one's donation, a nearly completed campaign may attract higher donations when presenting few rather than many donors involved.

In addition, and somewhat more intuitive, if there are only few (vs. many) donors to a nearly completed campaign, a potential donor may perceive that the norm for the average donation size is high, setting a higher baseline for their own donation, which may also increase donation amount (but not willingness to donate). In the current investigation, we test these two proposed accounts against the standard social norm hypothesis where observing many donors should increase motivation to give in the context of late-stage campaign, and find evidence for the former: information about few donors (vs. many) increases giving in late stages. Furthermore, two controlled experiments provide causal evidence suggesting that this effect is driven in part by an increase in the perceived progress that one's donation makes towards reaching the campaign goal rather than by adherence to a higher average donation size norm.

The current research makes important practical and theoretical contributions. Online engagement is becoming a critical component of a nonprofit organization fundraising strategy. While overall giving in the US grew 19% in the years of 2019–2021, online giving grew 42% over the same period [15], suggesting that traditional—offline—fundraising efforts are moving to the digital landscape. While individual contributions to online fundraisers play a significant role in human welfare and the alleviation of suffering, sadly, most fundraising campaigns globally do not reach their funding goals [15]. As such, behavioral science should expand the analysis of online giving and shed light on mechanisms that facilitate greater giving. Thus, our results have immediate implications for fundraisers because emphasizing the "right" information throughout the campaign lifecycle may have a significant impact on donations. This is especially true online where the environment is interactive and dynamic. We conclude the article by offering suggestions on how fundraisers can implement these findings to increase donations to their campaigns.

On the theoretical level, this research advances the focus theory of normative conduct [16–18] by illustrating for the first time how descriptive norm information can give rise to inferences about perceived goal progress—a key driver of donation decisions and goal-directed behavior more broadly. This work also provides a novel demonstration of when and why providing normative information can backfire. Unlike most previous research on the backfiring of social norms, or "boomerang" effects [19–21], we highlight situational conditions under which observing a *less* normative action can lead to *more* of that behavior, compared to when the action is more normative. Further, we illuminate a social component of goal-oriented motivation, whereby the behavior of others influences the degree to which perceived goal progress motivates goal-directed behavior.

## Social information in fundraising

When individuals receive social information about the donation behavior of others, they learn about descriptive norms. According to the focus theory of normative conduct, a distinction can be made between injunctive norms, which specify what individuals should do, and descriptive norms [16, 17]. Following past work drawing on this theory, we define descriptive norms as those that describe what other people do [16–18, 22, 23]. Research on descriptive norms indicates that a given behavior is viewed as more normative as the number of individuals who engage in that behavior increases [17, 24–26]. Furthermore, once learned about (or inferred), individuals often act in a way that is consistent with the norm, especially when it is not obvious what one should do [22]. Conformity with behaviors that are perceived as normative has been observed across many contexts, including voting [25, 27]; energy conservation [24, 28]; the cessation of littering [17]; gambling [29]; health behaviors such as alcohol use [30]; and, most relevant to the current investigation, charitable giving. For example, individuals donate more when they learn that others have previously donated a high (vs. low) amount [2, 4, 5].

One type of signal for a descriptive norm that may inform individuals about whether and how much to donate is the number of donors to a given fundraising campaign. In general, the extent to which people conform to norms increases as the number of individuals conveying the norms increases [14, 31]. Consistent with this, one investigation found that students were more likely to donate to charitable funds after learning that a large (vs. small) proportion of the student population had donated to these funds in the past [3]. Similarly, the frequency of donations to crowdfunding projects increases with the number of donors to those projects [32]. This research suggests that conveying to potential donors that many (vs. few) others have donated previously may be a reliable way to increase giving.

However, the relationship between number of contributors and giving may not be this straightforward. For example, research on the effect of group-size on contribution to public goods has found contradicting evidence and there is an ongoing debate whether cooperation increases or decreases in larger groups. Earlier works showed that compared to smaller groups, larger groups have a harder time providing a public good [33–35]. While scholars offer several explanations for the negative effect of group size on cooperation [36, 37], it has been proposed that when individuals who contemplate their contribution are part of a larger group, "the perceived impact of one's own contribution in determining collective outcomes is greatly reduced" [12]. In recent years, however, several studies showed that group-size has a positive effect on cooperation [38–40]. Finally, a meta-analysis by Zelmer [41] shows that the effect of group size on cooperation has not been consistent and does not seem to be strong. Research on the effect of group-size on contribution to public goods, therefore, offers no conclusive suggestion on whether and how information about the number of participants in a fundraising

campaign should affect motivation to give. Importantly, in the context of charitable giving or contribution to public goods, we are aware of no research examining the effect of social information on giving while taking into account goal proximity, an important factor that influences giving.

## Perceived progress and goal-directed behavior

When goals are approached, effort towards achieving those goals is increased. This finding, termed the goal gradient effect, was first documented in 1934 when Clark Hull observed that rats run faster as they get closer to reaching a food reward [6], though more recent research has demonstrated this effect in humans as well. For example, one study found that people who rate songs on a website in exchange for rewards visit the website more often, and rate more songs per visit, as they get closer to reaching the reward [7]. In addition, people make more frequent donations to fundraising campaigns when the campaign goal is near [9, 32, 42]. They also donate more when they believe that their donation will complete the campaign [43, 44]. Across many contexts, goals have been shown to exert a powerful influence on motivation [45–47].

One reason why people work harder as goals are approached is because late-stage actions result in an increase in the perceived impact one's actions have on progress toward the goal [7, 9, 11]. For instance, consumers made more frequent purchases to obtain a free coffee when they received a 12-stamp coffee card with two pre-existing bonus stamps, vs. a 10-stamp card with no pre-existing stamps [7]. This finding suggests that proportional distance to the goal influences motivation to complete goals to a greater extent than absolute distance to the goal. This psychological component of the goal gradient effect is also evidenced by work demonstrating that late-stage actions are given more blame (or credit) for an outcome, even when these actions logically have no greater influence than early-stage actions [48]. Similarly, individuals exhibit a greater willingness to contribute to a charity campaign that is close (vs. far) from reaching its goal because they perceive that their contribution will result in greater progress towards reaching the campaign goal [9]. Collectively, this evidence suggests that goal-directed motivation is fueled by the illusion that late-stage actions achieve greater progress towards goal completion [7, 49, 50].

## The negative effect of social information on giving

In this research, we challenge the standard effect of social information on giving, whereby more donors should increase motivation to give [3], and suggest that under some conditions this effect is less likely, and may even revere. Specifically, when a sufficient progress has been made towards the goal, willingness to donate may increase when individuals receive information about *few* rather than many donors to the campaign. As we later explain, we focus here on late-stage campaigns because learning that there are few donors to a campaign that is far from its goal may signal a higher uncertainty about the worthiness and/or the attainability of the campaign, thus countervailing the effect of the few donors information on giving. Learning that there are many donors to an early-stage campaign, however, may alleviate those uncertainties, thereby making the standard effect of social information on giving is more likely. In addition, motivation to give to early-stage campaigns is lower due to the goal gradient effect discussed above. Based on the discussion above, we hypothesize that:

*Hypothesis 1 (H1)*: When sufficient progress has been made towards the goal, learning that few (vs. many) people have donated to a campaign increases giving.

But why should learning about few (vs. many) donors increase motivation to give in late-stage campaigns? First, information about the number of donors and the percentage of campaign goal reached may lead to inferences about the average amount donated by others. Potential donors may then match to or "anchor" on the perceived average amount given previously [5, 51–54]. As such, holding goal proximity constant, a campaign with few (vs. many) donors sets a larger anchor (i.e., a larger perceived average donation) to which donors may adhere. Importantly, not only does this interpretation make no prediction about donation likelihood, but a higher perceived average donation can also hinder new contributions due to the higher threshold of giving (we thank an anonymous reviewer for this point). More formally:

*Hypothesis 2a (H2a)*: The effects of the number of donors on the donation amount (but not on the donation likelihood) to near-completed campaigns are driven in part by an increase in the perceived average donation by others.

A second, and presumably a less intuitive interpretation, is that descriptive norm information about the number of donors to a fundraising campaign can also increase the perceived impact of subsequent donations on goal completion. The fewer people that have donated to a campaign, the larger the percentage of goal progress accounted for by each donor on average. As such, an additional donation to a close-to-completed goal may be perceived to make a greater impact on goal progress when there are few donors. For example, the tenth donation to a campaign that reached 90% of its goal may perceived having a greater impact on the goal progress than the hundredth donation because responsibility for the progress reached is shared by fewer contributors. Put differently, despite the absolute distance (i.e., dollar amount) from the goal being unchanged by the number of donors, the greater proportional progress achieved by the few (vs. many) who have already donated may create the impression that one's own donation will result in greater goal progress. Consistent with this logic, a donor to a nearly-reached campaign goal might intuit along the lines of, "if the campaign has gotten this far from the contributions of just a few people, then my donation can really make a difference on the campaign goal." In this sense, observing that there are few donors when close to the goal is expected to strengthen the motivation to give (which will already be high due to the goal gradient effect) by enhancing the perceived progress that one's donation makes towards achieving the campaign goal. Conversely, in line with the abovementioned research on social loafing, the perceived progress is expected to diminish when there are *many* (vs. few) donors to nearly-completed campaigns, as a donor may perceive that her donation will have less impact on the progress towards the goal when she represents a smaller (vs. larger) proportion of the pool of donors (i.e., responsibility for the goal progress is shared by many other donors).

Importantly, we propose that an increase in the perceived progress one's donation makes on the campaign goal would in turn increase one's motivation to donate, which should be reflected in both the amount given and the probability of giving to the campaign. Thus, we predict that for late-stage fundraising campaigns with few (vs. many) past donors, an increase in perceived progress towards the campaign goal would increase both the amount given and the probability of giving to the campaigns.

*Hypothesis 2b (H2b)*: The effect of number of donors on donation amount and donation likelihood to near-completed campaigns is driven, at least in part, by an increase in the perceived progress one's donation makes towards goal completion.

If receiving information about few donors increases one's sense of progress when close to the campaign goal, wouldn't this also be true when the goal is far from being reached? We argue that this must not be the case. When only a small percentage of a campaign goal has been reached, people may interpret social information about the number of donors differently,

and countervailing forces may influence one's donation decision. Learning that there are few donors to a campaign that is far from its goal may send a signal about the worthiness or the attainability of the campaign. Because descriptive norms offer guidance about what to do in ambiguous situations [22], seeing that (a) not many people have donated and (b) little progress has been made may cast doubt on the perceived worthiness of the campaign's cause, as there is little social proof suggesting that donating is the correct thing to do. In addition, research on goal setting and task motivation has demonstrated that people are less motivated when goals seem less attainable [46, 55]. As such, having few donors and limited goal progress may signal to potential donors a higher uncertainty about whether the goal will be reached, thereby decreasing motivation to donate to the campaign. This notion is consistent with research showing that people are concerned with whether the goal is desirable or attainable when goals are far from completion, whereas they are concerned with making progress when goals are near completion [56]. For the reasons mentioned above, in this research we focus largely on the counterintuitive effects of social information on donation decisions when the campaign goal is near. However, we also expect that learning that a campaign with little goal progress and few (vs. many) donors would have no effect on giving, or possibly even decrease giving as the standard effect of social information on giving would predict.

*Hypothesis 3 (H3)*: When a campaign goal is far from being reached, learning that few (vs. many) people have donated has no effect or even decreases giving.

## Additional potential mechanisms

While we predict that donor's perception of their impact on the campaign progress and/or inferences about the average amount donated by others should underly the (negative) effect of social information on giving to nearly-completed goals, it is important to note that there could be other mechanisms driving this effect. It is possible, for example, that donors may derive emotional benefits, or "warm glow" [57–58], from becoming a part of a small (i.e., more exclusive) group of donors that is effective at advancing a campaign toward its goal. Furthermore, observing few donors to a near-complete campaign may provide a signal that the campaign's beneficiaries are in great need. We explore the potential roles of such mechanisms to shed light on why the counterintuitive negative effect of social information on giving might occur. Armed with knowledge about the underlying process, fundraisers can better design online interfaces for effective campaigns.

## General methods

We first analyze data from a real fundraising website to investigate whether the predicted negative effect of social information on giving to late-stage campaigns occurs. Then, in two experiments, we replicate this pattern and provide causal evidence for the proposed account.

## Ethics statement

All studies were approved (in writing) by the Human Research Protection Program of the University of California, San Diego (IRB number: 130572XX). Study 1 used data from the field and Studies 2 and 3 carried out at University of California, San Diego. Participants in Studies 2 and 3 were recruited from the Rady School of Management behavioral lab's subject pool and were undergraduate students who take part in laboratory studies for course credit. All participants in these studies provided written consent to participate. Study 1 data was provided directly by JGive with their permission to publish findings for research purposes and remained within their terms and conditions. Data, analysis scripts, and survey materials for Studies 2

and 3 are available on Open Science Framework (OSF) at https://osf.io/rb8s2. Study 1 data is proprietary and cannot be shared publicly.

## Study 1

In Study 1, we analyzed a large dataset from JGive (www.jgive.com), a platform that allows charities to conduct online fundraising campaigns. On JGive, prospective donors are shown up-to-date information about individual fundraising campaigns, including the percentage of the campaign goal reached and the current number of donors to the campaign. As such, data from this website allows us to test whether information about the number of donors can have a negative effect on giving in actual fundraising campaigns (H1). S1 Appendix includes detailed information about the data and the sampling procedure in Study 1.

### Dataset

The JGive dataset includes information about all individual donations that occurred on the website between November 2019 and February 2020. Each observation (i.e., individual donation) contains the donation amount in New Israeli Shekels (NIS), the time and date of the donation, and whether the donation was one-time or recurring. In addition, each observation includes campaign-level information such as a campaign ID and a charity ID (individual charities can run multiple campaigns), the goal amount, the number of unique donors to the campaign, the total amount collected after the donation was made, as well as whether the campaign offered a matching program in which large donors would match each donation with an equivalent amount. We excluded donations that were made after the goal had been reached, as well as those donations that were part of a recurring donation program because these donation decisions are unlikely to be influenced by the investigated factors. Our final dataset consisted of 42,702 observations representing 3,147 unique campaigns that were created by 143 charity organizations.

### Results

Summary statistics are displayed in Table 1. We used campaign-level information to calculate the number of donors and the corresponding goal proximity that was displayed to each donor before they made their donations. In addition, we included a dummy variable for whether the donation completed the campaign goal. We included this variable because research indicates that individuals are more likely to give when their donation completes the campaign goal [42–44, 59].

Regression results are shown in Table 2. In Model (1), we regressed the log of the donation amount (skewness = 52.12, *SE* = .012) on the number of past donors, the percentage of the goal reached, and their interaction. We note that seventy-seven charities (53.8%) in the JGive dataset initiated more than a single campaign. To account for this, we added random effects for the charity ID in our models as well as random effects for the campaign ID (nested within charity ID). As indicated in Table 1, Model (1) revealed an unpredicted negative effect of goal

**Table 1. Study 1, JGive data summary statistics.**

|  | Mean | St. Dev. | Min | Median | Max |
|---|---|---|---|---|---|
| Average donation | 250.65 | 856.35 | 3.43 | 100 | 103,796.4 |
| Percentage of goal reached | 26.63 | 26.33 | 0 | 18.56 | 99.9999 |
| Number of donors | 361.01 | 640.17 | 0 | 26 | 5,042 |
| Goal amount | 518,159 | 1,394,847 | 10 | 100,000 | 10,000,000 |

**Table 2. Study 1 regressions results.**

| | *Dependent variable*: Log of Average Donation Amount | |
|---|---|---|
| | (1) | (2) |
| Number of donors × Percentage of goal reached | -0.014* (0.007) | -0.014* (0.006) |
| Number of donors | 0.013 (0.016) | 0.052** (0.016) |
| Percentage of goal reached | -0.056*** (0.008) | -0.111*** (0.008) |
| Goal amount | | 0.039 (0.027) |
| Last donation | | 1.180*** (0.046) |
| Has matching | | 0.191* (0.078) |
| Campaign random effects | ✓ | ✓ |
| Charity random effects | ✓ | ✓ |
| Constant | 5.074*** (0.054) | 5.048*** (0.056) |
| N | 42,702 | 42,337 |
| BIC | 114,905 | 113,142 |

*Note*: All independent variables are standardized. Standard errors are presented in parentheses below parameter estimates. The dependent variable is transformed using the natural logarithm. Campaign random effects are nested within charity.

* $p < .05$

** $p < .01$

*** $p < .001$.

proximity such that as the campaign approached its goal, JGive donors gave smaller donations, $B = -.06$, $t(39,552) = 7.04$, $p < .001$. There was no significant main effect of the number of donors, $B = .01$, $t(39,552) = 0.76$, $p = .446$. Most importantly, however, we found evidence for a negative interaction between the number of past donors and the goal progress, $B = -.01$, $t(39,552) = 2.12$, $p = .034$. Model (2) is similar to Model (1) but includes several control variables: goal amount, whether the campaign had donation matching (yes = 1, no = 0), and whether the donation completed the campaign goal (yes = 1, no = 0). The negative interaction remained robust to the addition of control variables, $B = -.014$, $t(39,190) = 2.11$ $p = .035$. Thus, consistent with H1, when the campaign goal was close to being reached, campaigns with fewer donors received larger donations than those with many donors.

These results are displayed in Fig 1. This figure uses a predictive model in which goal amount was held at its median values. As predicted by the model, a campaign with an average goal that is 90% complete with four donors would expect a donation amount that is nearly NIS 40 (~$11) greater than a campaign with 1,723 donors at the same level of goal completion. In support of H3, the effect when far from the goal was in the opposite direction and smaller in size. For example, a campaign that is 10% complete with four donors would receive a donation that is around NIS 13 (~$4) less than a campaign with 1,723 donors at the same level of goal completion.

## Discussion

Using a large dataset from a real fundraising website, the results of Study 1 show that social information and goal proximity has an interactive effect on giving across campaigns. In support of H1, campaigns that were close to reaching their goals had higher average donations when those campaigns had few rather than many donors. In line with H3, when far from the goal, campaigns with many donors had a higher average donation, though the size of this effect was smaller compared to when close to the goal.

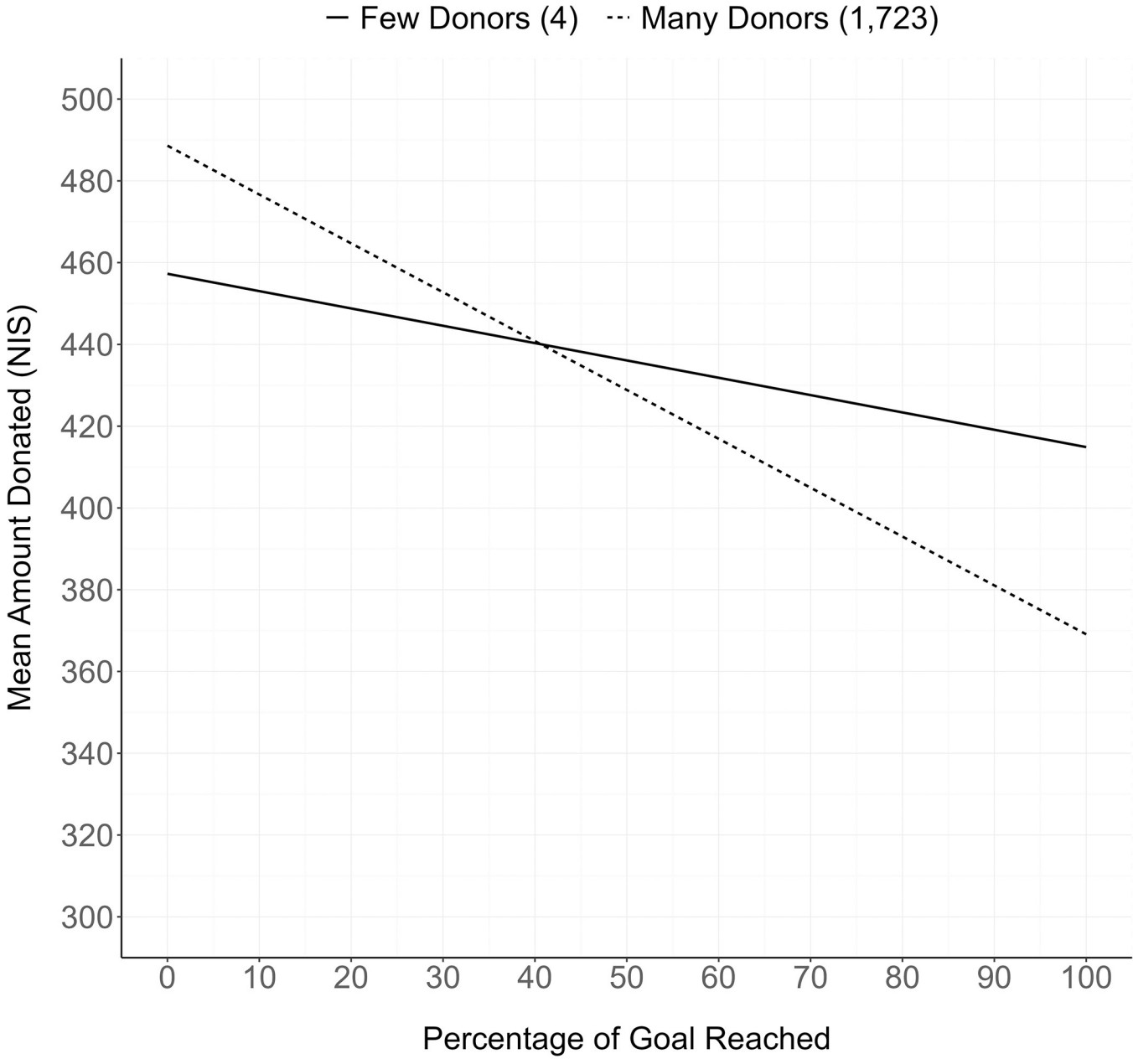

**Fig 1. Study 1: Predictive model illustrating the interaction between number of donors and percentage of the goal reached on average donation.**
Dependent value is not log-transformed for ease of interpretation of the predicted values (y-axis). We used 25th and 95th quantiles of the distribution of number of donors as "few" and "many" donors, respectively. Goal amount is held at its median value in both models.

It is interesting to note that in Study 1, we observed a negative effect of goal proximity information (in contrast to research on the goal gradient effect, e.g., [7]). We suggest two possible explanations for this result. First, given that a large proportion of the campaigns on JGive focus on local community needs, it is likely that members of those communities are among the first to learn about the fundraising endeavors. As such, larger donations from those who care about the cause the most are likely to be received at the early stage of JGive campaigns. Second, providing "seed" funding, or large initial contributions, to fundraising campaigns is an effective way to increase subsequent donations [60, 61]. On JGive, and unlike other crowdfunding

platforms, there is no restriction for project owners to contribute to their own project. Therefore, it is possible that creators of JGive campaigns "donate" large initial amounts to their own campaigns to attract more donors. Further research is necessary to better understand the circumstances under which the goal gradient effect does not hold.

Next, we ran controlled laboratory experiments to explore causal evidence for the observed effect and shed light on potential underlying mechanisms.

## Study 2

In Study 2, we experimentally manipulated the number of donors and the percentage of the goal reached to a real charity campaign in the laboratory and measured the resulting effects on giving to the campaign.

### Procedure and materials

We recruited participants from a university in southwest United States who completed the study in exchange for course credit and the opportunity to win $20. We received 571 complete responses and used all responses in our analysis ($M_{age}$ = 20.94, 57% female). We planned to collect as many responses as possible within the lab time that was allotted for the study.

In this experiment, participants were given the opportunity to donate to a real charity, Action Against Hunger (AAH). After learning about a specific charity campaign for AAH, the goal of which was to "fight high rates of chronic child malnutrition in Nicaragua," participants were randomly assigned to one of six conditions in a 3 (Number of Donors: few/many/no-information) x 2 (Goal Proximity: close/far) between-subjects design. Those in the *few* [*many*] conditions learned that there had been "23 [1,923] individual contributions" to the campaign, while those in the *no-information* control conditions received no information about the number of donors. In the *close* [*far*] conditions, a circle graph showed that the campaign was "86% [14%] funded."

Next, participants learned that they would be entered into a lottery for the chance to win $20 and that, if selected, they could donate all, some, or none of that money to AAH. All participants indicated how much of the $20, if any, they chose to donate in $1 increments. One participant was actually selected to receive the bonus based on her decision after data collection was complete, and we donated the amount she chose to AAH. Finally, participants completed manipulation check questions, additional exploratory follow-up items, and demographics (see S1 Appendix for more details; full survey materials are provided on OSF).

### Results

#### Manipulation check

As expected, those in the close conditions ($M$ = 5.07, $SD$ = 1.16) thought that the campaign was closer to reaching the goal than those in the far conditions ($M$ = 3.14, $SD$ = 1.25), $t(569)$ = 19.16, $p$ < .001, $d$ = 1.60 (1 = *very far*, 7 = *very close*). Additionally, those in the many conditions ($M$ = 4.38, $SD$ = 1.50) reported that there was a larger number of donors to the campaign than those in the few conditions ($M$ = 3.59, $SD$ = 1.53), $t(379)$ = 5.09, $p$ < .001, $d$ = 0.52 (1 = *a very small number*, 7 = *a very large number*). Those in the no-information control conditions reported an intermediate number of donors ($M$ = 4.11, $SD$ = 1.59). A post-hoc Tukey test revealed that the number of donors reported in the no-information control conditions was non-significantly different from that in the many conditions, $p$ = .13, and significantly greater than the number of donors reported by those in the *few* condition, $p$ < .01, $d$ = 0.33.

## Charitable giving

Overall, 86.7% of participants made a donation. The average amount donated (including those who donated nothing) was $12.70 ($SD$ = 7.49). A 3 x 2 ANOVA with number of donors and goal proximity entered as factors revealed no main effects on amount donated ($p$s > .60). On average, the amount given by those in the close conditions ($M$ = 12.78, $SD$ = 7.27) did not differ from that of those in the far condition ($M$ = 12.62, $SD$ = 7.72). Similarly, information about the number of donors did not influence the average amount given, with those in the few conditions ($M$ = 13.06, $SD$ = 7.52) giving a slightly higher amount than those in the many conditions ($M$ = 12.41, $SD$ = 7.52), and those in the control conditions giving an intermediate amount ($M$ = 12.63, $SD$ = 7.43). However, there was a significant interaction, $F(2, 565)$ = 3.94, $p$ = .02, $\eta^2_p$ = .01 (this effect also holds when controlling for gender and age, as well as when log-transforming the donation amount to account for non-normal distribution of standardized residuals; see S1 Appendix for this analysis). When the campaign goal was close to being reached, there was a significant effect of the number of donors on giving, $F(2, 284)$ = 3.53, $p$ = .03, $\eta^2_p$ = .02. Consistent with H1, those who saw that there were few donors ($M$ = 14.06, $SD$ = 6.99) gave more than those who saw that there were many donors ($M$ = 11.31, $SD$ = 7.69), $t(188)$ = 2.59, $p$ = .01, $d$ = .38. Those in the no-information control condition gave an intermediate amount ($M$ = 12.97, $SD$ = 6.91). A post-hoc Tukey test revealed that those in the no-information condition donated an amount that did not significantly differ from either the amount donated in the few or many conditions, $p$s > .20. When far from the goal, the effect of number of donors on giving was non-significant, $p$ = .38. These results are shown in Fig 2.

A similar pattern of results was observed for the probability of giving. A logit model with probability of giving as the dependent variable (1 = donated, 0 = did not donate) revealed no

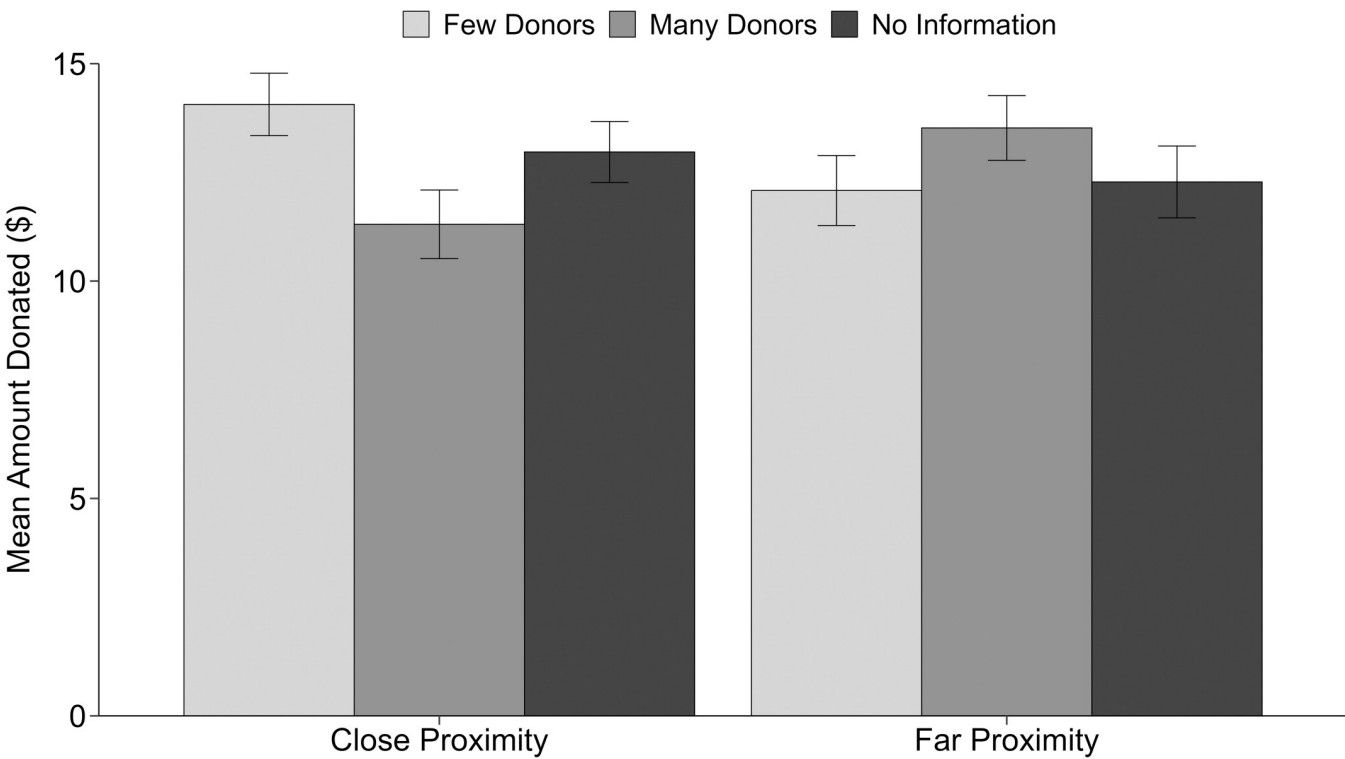

**Fig 2. Study 2 mean amount donated as a function of the number of donors (few/many/no information) and goal proximity (close/far) conditions.**

main effects ($ps > .10$) and a significant interaction, B = 1.31, z = 2.14, $p = .03$. When the goal was close to being reached, there was a marginally significant effect of number of donors, B = .73, z = 1.66, $p = .096$. Those in the few condition (90.5%) exhibited a marginally significantly higher probability of giving than those in the many condition (82.1%), $\chi2(1) = 2.85$, $p = .09$. Those in the no information control condition (91.8%) were as likely to give as those in the few condition, $p = .76$, but significantly more likely to give than those in the many condition, $\chi2(1) = 3.94$, $p < .05$. When the goal was far from being reached, participants in the many conditions (89.4%) were slightly more likely to give than those in the few condition (82.4%), but this difference did not reach significance, $p = .17$ (thus supporting H3). The intermediate probability of giving of participants in the control condition (83.9%) did not differ from either those in the few condition or those in the many condition, $ps > .20$.

## Discussion

In Study 2, we obtained causal evidence for the negative effect of social information on giving to nearly completed goals as predicted by H1. When a campaign goal was close to being reached, participants donated significantly more, and demonstrated a marginally greater likelihood of giving, when they learned that there were few donors to the campaign compared to when there were many donors.

## Study 3

Building on previous results, Study 3 examined the role of perceived progress and other potential mechanisms for the negative effect of social information on giving to late-stage campaigns. As described earlier, learning that a campaign goal has almost been completed by a small (vs. large) number of individuals may boost giving due to an increase in the perceived average amount donated by others (H2a) or by signaling that one's donation results in greater progress towards reaching the goal (H2b). In Study 3, we evaluated these and other potential mechanisms.

This study also tested the robustness of the effect in light of a potential boundary condition: whether the campaign focused on accumulated vs. remaining goal progress. This test is important given that some charity campaigns display accumulated goal progress, while others display remaining goal progress. Research on motivation and goal pursuit suggests that when commitment to a goal is weak (e.g., when asked to make a first-time donation), focusing on progress that has already been accumulated (vs. the progress remaining to goal completion) signals that a goal is desirable and feasible. This increase in perceived desirability and feasibility of the goal in turn increases motivation to achieve that goal [62]. Given that participants in Study 2 were exposed to the fundraising campaigns in the lab and did not seek this information on their own, they likely did not have a strong commitment to the campaign goals. Moreover, in Study 2, participants viewed a campaign that displayed only accumulated progress. It is possible, then, that when a campaign focuses on the goal progress remaining (vs. accumulated), a decrease in the desirability and feasibility of goal achievement would overshadow the uncovered effect. If this effect is contingent on the campaign displaying accumulated goal progress, then we would expect the effect to be eliminated or reversed when the focus shifts towards remaining goal progress.

### Procedure and materials

Seven hundred and nine participants from a university in the southwest United States completed our study in exchange for course credit and an opportunity to win up to $20 based on their donation decision as before ($M_{age} = 21.25$, 48% female). Participants in Study 3 viewed the same charity campaign and past donors manipulation that were given in Study 2, except all

participants saw that the campaign was close to reaching its goal. In this 2 (Number of Donors: few/many) x 2 (Progress Focus: accumulated/remaining) between-subjects design, those randomly assigned to the *accumulated progress* conditions received the same stimuli from the close condition in Study 2 (i.e., a circle graph indicating that the campaign was "86% funded"). For the *remaining progress* conditions, participants viewed a circle graph indicating that the campaign goal had "14% remaining" (see S1 Appendix for study materials).

Next, participants learned about the opportunity to win $20 and donate some, none, or all of this bonus to the charity. They then made their donation decision with the same prompt and $0–20 scale used in Study 2. One participant was randomly selected after data collection to win the money based on her decision and we donated the amount she chose to the charity.

### Exploring potential mechanisms

Following the donation decision, we collected several measures that may explain the negative effect of number of donors on giving when campaigns are close to their goals. The *perceived average donation* was measured with the item, "To the best of your understanding, how large or small is the average amount donated in this campaign?" (1 = *very small*, 7 = *very large*). We measured the *perceived progress* that one's donation makes towards the campaign goal with three items (1 = *not at all/none at all*, 7 = *a great deal/very much*), which were adapted from Cryder et al. [9] (e.g., "How much progress would an additional donation make toward the campaign goal?"; α = .89; Though Cryder et al. [9] label this construct as "impact," we use the term perceived progress to avoid conflation with other possible meanings of impact, such as the perceived magnitude of benefits to beneficiaries of the charity). We also measured the *perceived importance* of the campaign (1 = *not at all important*, 7 = *very important*; 2 items; e.g., "How important is it for you to help the campaign reach its goal?"; α = .84) and the *perceived importance others place on campaign* (2 items; e.g., "How important do you think it is for others to help the campaign reach its goal?"; α = .71). Additionally, we asked about *anticipated positive affect*, or warm glow [57, 58] and *anticipated negative affect*, respectively, with the following items: "How good would you feel if you helped the charity"; "How bad would you feel if you did not help the charity?" (1 = *not at all*, 7 = *very much*).

Furthermore, we included questions about the perceived *likelihood of goal completion without one's donation* ("To the best of your understanding, how likely is it that the campaign goal will be reached without your donation?"; 1 = *very unlikely*, 7 = *very likely*), *perceived need* [63, 64] ("To what extent do you feel that the people the charity helps are in need?"; 1 = *not at all*, 7 = *very much so*), and feelings of *moral responsibility* to help [65] ("How much do you feel it is the right thing to do to help out with the cause?"; 1 = *not at all*, 7 = *very much so*). Finally, participants responded to a manipulation check question that was similar to that in Study 2 (full survey materials are provided on OSF).

## Results

### Manipulation check

As expected, those in the many conditions ($M$ = 4.76, $SD$ = 1.53) reported that there was a larger number of donors to the campaign than those in the few conditions ($M$ = 4.28, $SD$ = 1.46), $t(707)$ = 4.30, $p < .001$, $d$ = .32.

### Charitable giving

Overall, 90.0% of participants donated to the charity, with an average donation of $12.89 ($SD$ = 7.25), including those who did not donate. A 2 (number of donors: few/many) x 2

(progress focus: accumulated/remaining) ANOVA revealed a main effect of number of donors on the amount donated, $F(1, 705) = 8.20$, $p < .01$, $\eta^2_p = .01$. As expected, those in the few conditions ($M = 13.66$, $SD = 7.05$) donated more than those in the many conditions ($M = 12.11$, $SD = 7.36$), $t(707) = 2.87$, $p < .01$, $d = .21$ (thus supporting H1). A post-hoc Tukey test confirmed this result ($p = .004$). There was no effect of the progress focus manipulation and no interaction ($ps > .80$).

We also examined the effects of our experimental treatments on the likelihood of donating using a logit model with a binary variable indicating whether the participant donated entered as the dependent variable (1 = donated, 0 = did not donate), and number of donors, progress focus, and their interaction entered as independent variables. There was a marginally significant effect of number of donors, $\beta = -.06$, $z = -1.74$, $p = .08$, such that those in the few conditions (92.1%) had a marginally greater likelihood of donating than those in the many conditions (87.9%), $\chi2(1) = 3.57$, $p = .06$. As before, there was no effect of progress focus and no interaction, $ps > .50$. We therefore collapsed the data across progress focus in all subsequent analyses.

### Potential mechanisms

To assess mechanism, we ran a multiple mediation model that included the number of donors as the independent variable and amount donated as the dependent variable (PROCESS Macro for SPSS, Model 4 [66]). All measured potential mechanisms (perceived average donation; perceived progress; importance placed on campaign; perceived importance others place on campaign; anticipated positive affect; anticipated negative affect; likelihood of goal completion without one's donation; perceived need; and moral responsibility) were entered as mediators simultaneously in the model. Among all potential mechanisms, only perceived progress had a significant indirect effect on the donation amount, $\beta = 0.21$, 95% CI [0.04, 0.46] (thereby supporting H2b). Observing few (vs. many) donors increased participants' sense of perceived progress, $\beta = 0.29$, $p = .01$, which in turn drove an increase in the amount donated, $\beta = 1.65$, $p < .001$. Confidence intervals for all other potential mediators contained zero.

In addition to testing mechanisms behind the effect of number of donors on the amount given to the campaign, we examined the mechanisms underlying this effect on the probability of giving to the campaign. For this analysis, we used the same mediation model as described above (collapsing across progress focus), except that the probability of giving was used as the dependent variable, rather than the amount given. This test revealed virtually identical results: There was a significant indirect effect of progress, $\beta = 0.22$, 95% CI [0.04, 0.54] (b path between mediator and dependent variable: $\beta = 0.07$, $p < .001$), and no other significant indirect effects (confidence intervals for all other potential mediators contained zero). Indirect effects for all items in this model and the continuous dependent variable model are displayed in S1 Appendix.

### Discussion

The results of Study 3 replicated those of the previous studies and highlighted a potential mechanism as predicted by H2b. That is, observing a small (vs. large) number of donors in a late stage of fundraising efforts increased the sense of progress that participants' donations made towards the campaign goal. Furthermore, the results of Study 3 indicate that this effect holds regardless of whether the campaign focuses on progress accumulated thus far or the progress remaining to goal completion.

### General discussion

This paper provides evidence that individuals give more to nearly completed campaigns when they observe that few (vs. many) people have donated. We also find evidence that this effect is

driven, at least in part, by the perception that one's donation results in greater progress towards reaching the campaign goal. These findings suggest that managers of fundraising institutions who are currently displaying both number of past donors and goal proximity to potential donors should reconsider their strategy, as attempts to convey social proof through the number of previous donors may sometimes backfire. Moreover, the current investigation expands our understanding of the interplay between descriptive norms, goal proximity, and giving at a theoretical level.

This research has several important implications for fundraisers. Natural variation in the number of donors to campaigns poses a challenge to fundraisers seeking to put these findings into practice. However, this problem is not insurmountable. One potential strategy is that when the campaign goal is near, fundraisers should draw prospective donors' attention away from the number of donors if this number is relatively large. Conversely, they could emphasize information about donors when they are relatively few in number by making that information more visually salient. Another possible strategy is to run multiple campaigns with smaller goals rather than one campaign with a single large goal, as modest goals are more likely to be completed by a small number of donors. This strategy might benefit even further from implementing a large seed donation [60, 61] that produces substantial goal progress. More studies, however, should directly test this strategy in real fundraising campaigns. For nonprofit organizations, our findings may also emphasize the importance of cultivating a pool of major donors. If nonprofits can obtain large donations from these individuals and thus make significant progress towards the campaign goal with relatively few donors, not only does this allow for faster progress toward the goal, but it can also attract additional donations by showing that this progress has been made by a relatively small number of donors. We encourage fundraisers to test these strategies, as they could help bring in money for worthy causes that might otherwise be lost.

Our investigation also advances theory in several ways. One theoretical contribution is that this research highlights how descriptive norm information can lead to inferences about perceived goal progress. Past research shows that one explanation for why social information positively influences giving [2–4] is that the information may signal that giving is the right thing to do [67]. For instance, the campaign could be seen as high quality or benefitting a worthy cause, or people might infer that they can meet real or imagined affiliation goals with others by donating. However, in this work, we establish that when a goal is close to being reached, social information about the number of donors can in fact lead to the belief that one's donation can have a greater impact on progress towards reaching the goal. This is an important result because perceived goal progress is a key driver of charitable giving [9] and also plays a critical role in goal-directed motivation more broadly [7, 11]. To our knowledge, this is the first demonstration that social information can affect inferences about goal progress.

Additionally, this work identifies novel conditions under which the effects of descriptive norms backfire, which are doubly important given the popularity of appeals to descriptive norms in marketing efforts. Backfiring of the intended effects of normative information, or a "boomerang effect," has been documented in research showing that the use of descriptive norms without injunctive norms can lead to bad behavior when the descriptive norm details the prevalence of the bad behavior [20, 21]. However, our work differs from this and other research highlighting moderators that increase or decrease the effectiveness of descriptive norms [18, 21]. Here, we show that presentation of a less normative action (i.e., few donors) can lead to more of that behavior than when the action is portrayed as more normative (i.e., many donors). As this research demonstrates, in some contexts, less (vs. more) normative behavior can be the more potent catalyst of conformity.

This work also opens up several other potential areas for future research. First, it is important to test the effects of social information and goal proximity in other contexts with different types of goals as well as with different kinds of social information. Furthermore, subsequent research could help examine moderators of this effect. For instance, one key variable that could moderate this effect is the degree to which individuals are committed to a goal. It is possible that for nearly completed goals to which one is strongly committed, observing that there are few contributors could further galvanize motivation to achieve the goal, as progressing the goal may be of greater personal importance to that individual. Another potential moderator is the extent to which individuals identify with contributors to the goal. Past work has shown that descriptive norms influence behavior more when individuals share an identity with those whose behavior they learn about or observe [68–70]. Thus, it is possible that when the social information is related to people with whom one closely identifies, the desire to conform with these individuals may lead to a stronger effect of norms. More research is necessary to test these notions.

Other alternative explanations to the observed effect are also plausible as many behavioral effects are multi-determined. While Study 3 finds perceived impact on the campaign goal to mediate the effect, more nuanced psychological factors may better explain changes in this perception. For example, the collective efficacy hypothesis assumes that in the context of group efforts, people motivate and guide their actions by the group's common belief in its joint ability. Shared beliefs in the collective efficacy influence how much effort people put into their group endeavor [55, 71] and a high sense of collective efficacy is found to promote social behaviors such as volunteering and money donations [72]. It is thus possible that in the context of fundraising, observing a large amount of goal progress being achieved by a relatively small number of donors increases the sense of collective efficacy which in turn motivates people to exert more efforts in helping the campaign reach its fundraising goal. Future research may explore the role of perceived collective efficacy more closely.

As with most behavioral studies, the current study is not without limitations. We note that both lab studies draw from a student population and the relative high donation rate reported in these studies may not generalize to other fundraising contexts. While future studies may also test these effects in other populations, we note that Study 1 includes data from real donors (i.e., not student participants). It is possible that the donation likelihood observed in the lab studies was high partly because students were offered to donate from an additional bonus which was on top of the expected course credit they received for their participation (i.e., windfall). In addition, most participants did not donate the entire bonus but instead kept some portion of the bonus for themselves.

## Conclusions

As social animals, we often look to the actions of those around us for cues to how we should behave. Often, this results in people conforming with the behavior of others. As we show in this research, however, other pieces of information in our environment can change how we interpret and respond to social information. In the context of fundraising, learning that a campaign goal has been nearly completed by few individuals can be a more powerful motivator of subsequent giving than when many individuals have accomplished that same level of goal progress. While this work presents a counterintuitive effect of social information in the presence of goal proximity information, we call on further research on this topic to better understand the interplay between social information and goals under different contexts, as well as to uncover new and effective ways to nudge people to give.

## Supporting information

**S1 Appendix. Detailed information about Study 1 data and the sampling procedure as well as Studies 2 and 3 stimuli and additional results.**
(DOCX)

## Author Contributions

**Conceptualization:** Coby Morvinski, Matthew J. Lupoli, On Amir.

**Data curation:** Coby Morvinski, Matthew J. Lupoli.

**Formal analysis:** Coby Morvinski, Matthew J. Lupoli, On Amir.

**Investigation:** Coby Morvinski, Matthew J. Lupoli, On Amir.

**Project administration:** Coby Morvinski, Matthew J. Lupoli, On Amir.

**Supervision:** Coby Morvinski, Matthew J. Lupoli, On Amir.

**Validation:** Coby Morvinski, Matthew J. Lupoli, On Amir.

**Visualization:** Coby Morvinski, Matthew J. Lupoli.

**Writing – original draft:** Coby Morvinski, Matthew J. Lupoli, On Amir.

**Writing – review & editing:** Coby Morvinski, Matthew J. Lupoli, On Amir.

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
