## [Decision Letter · Decision Letter 0]

22 Apr 2022

PONE-D-22-05379Social Information Decreases Giving in Late-stage Fundraising CampaignsPLOS ONE

Dear Dr. Morvinski,

Thank you for submitting your manuscript to PLOS ONE. After careful consideration, we feel that it has merit but does not fully meet PLOS ONE’s publication criteria as it currently stands. Therefore, we invite you to submit a revised version of the manuscript that addresses the points raised during the review process.

I received two reviews from experts in the field.  Both reviewers note major issues with your manuscript that preclude publication.  Reviewer 2 is more positive than Reviewer 1, who argues for rejection.  If you feel you can adequately address the reviewers' concerns, then I will allow you to submit a revised manuscript.  If you decide to revise the paper, then I will send it back to the original reviewers.  Note that the reviewers raised some serious concerns about the manuscript.  Your manuscript would need to be extensively improved in order to convince the reviewers that your conclusions are supported by the data.

We look forward to receiving your revised manuscript.

Kind regards,

Darrell A. Worthy, Ph.D

Academic Editor

PLOS ONE

Journal Requirements:

2. PLOS ONE has specific requirements for studies using personal data from third-party sources, including social media, blogs, other internet sources, and phone companies (https://journals.plos.org/plosone/s/submission-guidelines#loc-personal-data-from-third-party-sources). These requirements include confirming data are collected and used in accordance with the company or website’s Terms and Conditions, obtaining appropriate ethics or data protection body review, and ensuring appropriate consent from individuals whose data are used in research. In this case, please ensure that your Ethics statement is in compliance with guidelines, and that you have complied with the company's (i.e., Kickstarter and JGive's) Terms and Conditions, with appropriate permissions.

6. Please include a caption for figure 2.

Reviewers' comments:

Reviewer's Responses to Questions

**Comments to the Author**

1. Is the manuscript technically sound, and do the data support the conclusions?

Reviewer #1: No

Reviewer #2: Partly

2. Has the statistical analysis been performed appropriately and rigorously? 

Reviewer #1: No

Reviewer #2: Yes

3. Have the authors made all data underlying the findings in their manuscript fully available?

Reviewer #1: No

Reviewer #2: No

4. Is the manuscript presented in an intelligible fashion and written in standard English?

Reviewer #1: Yes

Reviewer #2: Yes

5. Review Comments to the Author

Reviewer #1: Study 1a: The authors use snapshot data from different campaigns. I do not understand how the authors arrive at their conclusions but maybe I do not understand the structure of the data correctly. Do the authors have the information about the last donation? Then, one could analyze the size of the last recorded donation conditional on the completion rate, number of previous donors, and average donation of previous donors. However, I think that the authors do not look at the incremental but at the average donation. In such a case the results are a kind of tautology. Think of two identical campaigns, A and B, with the same goal and exactly the same completion rate. If B has received more donations then, necessarily, the average donation in B must be lower. If this is a case, I do not see any value in presenting this data.

Minor:

Table 2: N is missing

Study 1b seems to be doing what I suggested above, however, I am not sure that the model is sufficiently taking care of the dynamics of the data. I wonder whether campaigns with many donors that result in many observations in the data, receive a too high weight, and whether this affects the coefficients. Does using weights [1/N] change the regression results? Actually, the authors have sufficient data to run separate (between projects) regressions, say using the incremental new donation amount after 70% of the goal has been reached (80%, 90%, etc). Then one could regress the new donation on the number of previous donations and the amount needed for completion. In the original regression, and the given setting, I am also worried about the correlation between the number of donors and the percentage of goal reached. Within one project there must be a very high correlation such that I am not sure how to interpret the coefficients. This would be taken care of in the regression proposed above.

Study 2: I think that the authors are overlooking a standard public good problem for which we know that with increasing N the average contribution is declining. The other issue is that of pivotality. If there is 86% of the goal reached and only 23 individuals donated previously that it is more likely that I am pivotal to finish the project while with the same amount reached by almost 2,000 previous donors, my donation is really negligible. (Strangely, there seem to be no information about the goal amount, but the goal amount must be expected to be larger with almost past 2,000 donations than with 23 past donations).

Procedures: I am very worried about the deception in the material that the authors present to the subjects: where did they take the numbers 23/1923/86%/14% from? Similarly, were the participants aware of the probability of their decisions being implemented? I consider the probability of 1/571 to be really negligible for individual decisions. The authors should have rather chosen fixed probability of 1/10 or 1/20.

Study 3 Introduction: I am confused when the authors talk about study 2 and 3 that “participants viewed … only accumulated progress.” Was Study 3 not aiming at changing this?

My remarks to the procedures are the same as those to study 2.

Do the authors correct for multiple hypothesis testing?

Implications:

I am very worried that the policy implications which the authors propose are not sufficiently backed up by the studies conducted. If the authors suggest that charities might split a large goal into smaller goals that could be achieved in smaller groups, why do not test this directly.

Overall, I am worried that the findings are quite trivial and in line with basis economic models of public good contributions and related to pivotality. Of course, one can test simple theory but the authors should spell it out at the beginning. If the results are not with the simple theory then it would be good to search for other—behavioral—explanations. I suggest to drop Study 1a completely and move study 1b to the Appendix.

In addition, there are some papers on the completion effect in crowdfunding campaigns that the authors should include in their paper:

Argo, Nichole, et al. "The completion effect in charitable crowdfunding." Journal of Economic Behavior & Organization 172 (2020): 17-32.

Cox, J., Tosatto, J. and Nguyen, T. (2022), "For love or money? The effect of deadline proximity on completion contributions in online crowdfunding", International Journal of Entrepreneurial Behavior & Research, Vol. ahead-of-print No. ahead-of-print. https://doi.org/10.1108/IJEBR-04-2021-0317

I believe there are more.

In top, the dynamics of crowdfunding campaigns are studied in several papers by Cason and Zubrickas.

Reviewer #2: Dear authors,

I am pleased to review your very interesting article about the interaction between number of donors and reaching funding goal on average donation amount. I think you are studying a very interesting, counterintuitive effect that might help us to understand the boundary conditions giving. Also, I highly value the overall study design, combining secondary, longitudinal data with experiments, as well as replicating findings in a different context (Study 1a and Study 1b). I am happy to provide suggestions to improve the manuscript.

1) Data and syntax availability. Given recent open science practices, I would have expected to have access to data and syntax of the work. I understand that there will be an OSF repository after acceptance, and that some data (Study 1b) is proprietary. Still, as a reviewer, I would appreciate if I could receive a (anonymous) link to the data.

2) Introducing the scope of the project. The introduction might make clearer, at the beginning, why the two variables under study (and their interaction effect) are so important. Instead of presenting it as a possible counterintuitive effect, I would like to see some more urgency and rationale. Do you have empirical examples that hint at this effect? (so an interesting empirical puzzle, as an anomaly)? Or is the current literature missing/lacking explanations for a certain phenomenon. I found the third paragraph (starting with however) particularly confusing. The mechanism of the hypothesis remains unclear, and I also noticed that you write about the likelihood of giving, whereas you use the log of the average donation amount in the studies.

3) Theoretical grounding. While the empirical approach is sound, and methods are described in a clear matter, I think the manuscript can improve in terms of theoretical grounding. I think the main point is that I could not understand whether it is a deductive or inductive paper. On one hand, it is deductive since you propose a central, interaction hypothesis in the introduction. On the other hand, the paper lacks a conceptual model and hypotheses (except the brief part in the introduction). But, even when it is more inductive, I would have expected more grounding in prosocial and fundraising literature. I would suggest to provide more argumentation in the introductory parts of the studies. Last, I think that the discussion could be improved by elaborating on the theoretical implications, and possible alternative explanations (e.g. I had to think about Bandura's collective efficacy as a potential mechanism). The contributions to the prosocial literature remain unclear.

4) Methods. First, I was wondering whether the Kickstarter case fits your RQ and scope. I have participated in several Kickstarter projects, and while there might be some prosocial motivations, it is often more of an investment (with the promise of a product/service) than a donation. Hence, the empirical context of Kickstarter is somehow disconnected from the RQ, and hence I am not surprised that the findings between Study 1a and 1B differ. Second, I noticed you worked with samples of students for your experiments. It might be possible to use students for experiments, but it is common practice in Psychology to explain why students would be a suitable population to study your RQ. Third, while I actually liked that the experiment included a real cause (increasing the psychological realism), I was surprised by the high percentage of respondents giving money. This might be an important limitation: does the empirical approach (experiment) somehow put pressure to give (a small amount of) money to the cause, making the behavior less realistic?

I hope my suggestions and questions help the authors to further improve their manuscript.

6. PLOS authors have the option to publish the peer review history of their article (what does this mean?). If published, this will include your full peer review and any attached files.

Reviewer #1: No

Reviewer #2: No

---

## [Author Response · Author response to Decision Letter 0]

22 Jul 2022

Please find enclosed a response letter.

---

## [Decision Letter · Decision Letter 1]

9 Sep 2022

PONE-D-22-05379R1Social Information Decreases Giving in Late-stage Fundraising CampaignsPLOS ONE

Dear Dr. Morvinski,

Thank you for submitting your manuscript to PLOS ONE. After careful consideration, we feel that it has merit but does not fully meet PLOS ONE’s publication criteria as it currently stands. Therefore, we invite you to submit a revised version of the manuscript that addresses the points raised during the review process.

I sent your paper back to one of the original reviewers, and the reviewer still lists several concerns.  If you feel you can address these concerns then I invite you to submit a revision.  If the reviewer still has major concerns after the next round of review then I will likely reject the paper, so please give your best effort to adequately address the remaining concerns raised by the reviewer.

We look forward to receiving your revised manuscript.

Kind regards,

Darrell A. Worthy, Ph.D

Academic Editor

PLOS ONE

Reviewers' comments:

Reviewer's Responses to Questions

**Comments to the Author**

1. If the authors have adequately addressed your comments raised in a previous round of review and you feel that this manuscript is now acceptable for publication, you may indicate that here to bypass the “Comments to the Author” section, enter your conflict of interest statement in the “Confidential to Editor” section, and submit your "Accept" recommendation.

Reviewer #2: (No Response)

2. Is the manuscript technically sound, and do the data support the conclusions?

Reviewer #2: Partly

3. Has the statistical analysis been performed appropriately and rigorously? 

Reviewer #2: Yes

4. Have the authors made all data underlying the findings in their manuscript fully available?

Reviewer #2: Yes

5. Is the manuscript presented in an intelligible fashion and written in standard English?

Reviewer #2: Yes

6. Review Comments to the Author

Reviewer #2: Thank you for the opportunity to review the revision of your interesting work. Overall, I applaud the authors for the amount of work that went into this revision. The authors really engage with the suggestoins and questions of the previous round. Still, I have some remaining concerns.

- I appreciate that the data is partly available. I miss documentation (e.g. metadata) to understand the dataset properly.

- While the authors has put a lot of work in rewriting the introduction, I still think that the rationale and argumentation is convoluted. I don’t understand the mechanism “Similarly, giving to a nearly completed campaign with few (versus many) donors may signal that one’s donation will make a greater marginal impact on goal progress, which consequently increases motivation to give”, since the gap to be bridged to the end goal is the same for campaigns with many and with few donors. Alternatively, could it not be that if there are only few donors (but the end goal is almost reached) that a potential donor estimates the average norm for a donation is high, setting a baseline for his/her own donation? Counterintuitively, you would expect that the likelihood of giving would go down, because the threshold of giving is higher due to this descriptive norm.

- The new paragraph on theoretical and practical contributions is very generic, and should be more precise in how this paper contributes, and why this contribution is so urgent to understand.

- I appreciate the further elaboration of the theory section. However, I found the hypothesis development quite unclear and ungrounded. First, the argumentation for the specific hypotheses is not backed with literature. The paragraph starting with “In this research, we hypothesize that descriptive norm information about the number of donors to a fundraising campaign can also increase perceived progress towards goal completion…” on p. 9 does not refer to research. So, the reasoning for a higher motivation towards an end-goal is clear, and the reasoning for a positive effect of fewer previous donors on giving is clear, but the interaction between the two remains ungrounded. Second, I would split the argumentation for hypotheses 1 and 2.

- The paragraphs about alternative mechanisms in the theory section confuses the reader. It might be better to move and integrate this in the discussion.

- Thank you for further explaining the methodological details. I appreciate your decision to remove the Kickstarter study. Still, I question the ecological validity of the experiments after reading your explanation of the specific population and the incentive/bonus.

7. PLOS authors have the option to publish the peer review history of their article (what does this mean?). If published, this will include your full peer review and any attached files.

Reviewer #2: No

---

## [Author Response · Author response to Decision Letter 1]

7 Nov 2022

Please see attached response letter

---

## [Editor Report · Decision Letter 2]

16 Nov 2022

Social Information Decreases Giving in Late-stage Fundraising Campaigns

PONE-D-22-05379R2

Dear Dr. Morvinski,

We’re pleased to inform you that your manuscript has been judged scientifically suitable for publication and will be formally accepted for publication once it meets all outstanding technical requirements.

Kind regards,

Darrell A. Worthy, Ph.D

Academic Editor

PLOS ONE
---

## [Editor Report · Acceptance letter]

21 Nov 2022

PONE-D-22-05379R2 

Social Information Decreases Giving in Late-stage Fundraising Campaigns 

Dear Dr. Morvinski:

I'm pleased to inform you that your manuscript has been deemed suitable for publication in PLOS ONE. Congratulations! Your manuscript is now with our production department. 

Kind regards, 

on behalf of

Dr. Darrell A. Worthy 

Academic Editor

PLOS ONE